# ARCHITECTURE AGNOSTIC NEURAL NETWORKS

## ABSTRACT

In this paper, we explore an alternate method for synthesizing neural network architectures, inspired by the brain's stochastic synaptic pruning. During a person's lifetime, numerous distinct neuronal architectures are responsible for performing the same tasks. This indicates that biological neural networks are, to some degree, architecture agnostic. However, artificial networks rely on their fine-tuned weights and hand-crafted architectures for their remarkable performance. This contrast begs the question: Can we build artificial architecture agnostic neural networks? To ground this study we utilize sparse, binary neural networks that parallel the brain's circuits. Within this sparse, binary paradigm we sample many binary architectures to create families of architecture agnostic neural networks not trained via backpropagation. These high-performing network families share the same sparsity, distribution of binary weights, and succeed in both static and dynamic tasks. In summation, we create an architecture manifold search procedure to discover families of architecture agnostic neural networks.

## 1    INTRODUCTION

Fascinated by the developmental algorithms and stochasticity inherent in the developmental synaptic pruning process, in this paper, we will explore architecure agnostic neural networks via the lens of binary, sparse, networks. We ground our study using sparse binary neural networks because these networks capture many of the most salient aspects of biological networks:

- distinct neuronal units implementing non-linear functions in constrain an output to (-1, +1)
- synaptic connections that are restricted to (-1, +1)
- inhibatory and excitatory connections are represented by (-1, +1) respectively

In this paper we demonstrate that (i) AANNs exist in silico, (ii) high-performance sparse binary neural networks on static (MNIST classification) and dynamic (imitation learning on car-racing) tasks exist, and (iii) that our $\underline{s}$tochastic $\underline{s}$earch a$\underline{n}$d $\underline{s}$ucc$\underline{e}$ed (SENSE) algorithm explores the architecture manifold.

## 2    RELATED WORK

Biological neural networks endow organisms with the ability to perform a multitude of tasks, ranging from sensory processing (Glickfeld & Olsen, 2017; Peirce, 2015), to memory storage and retrieval (Tan et al., 2017; Denny et al., 2017), to decision making (Hanks & Summerfield, 2017; Padoa-Schioppa & Conen, 2017). Remarkably, these complex tasks persist throughout our lives despite neuronal pruning, and synapse deletion up until adulthood. This partially stochastic process of neuronal refinement is known as developmental synaptic pruning.

Developmental synaptic pruning occurs when the physical connection between a neuron's dendrite and another neuron's axon is eliminated (Riccomagno & Kolodkin, 2015), preventing any further relay of information. Interestingly, between infancy and adulthood mammals lose roughly 50% of their neuronal synapses (Chechik et al., 1999). A study in humans estimated that our prefrontal cortex dendritic spine density, a proxy for synaptic density, is on average more than two times higher in childhood than adulthood (Petanjek et al., 2011). This evolved process is also partially stochastic (Vogt, 2015). One of the main manifestations of stochastic developmental variation in the brain

occurs at the circuit level (Clarke, 2012), insinuating that there are many similar neural architectures that would have sufficed in place of your current brain's architecture! Given the ubiquity, extent, and stochastic nature of developmental synaptic pruning there are many theories for why this process exists: to increase information transfer efficiency (Horn et al., 1998), or to derive optimal synaptic architectures (Chechik et al., 1999).

Previous work in the machine learning field has sought out several methodologies to search the architecture manifold. Neural architecture search methods enabled traversing the architecture space to discover high-performance networks, by making them malleable to neuro-evolution strategies (Stanley & Miikkulainen, 2002; Real et al., 2017; 2018), reinforcement learning (Zoph & Le, 2016) and multi-objective searches (Elsken et al., 2018; Zhou & Diamos, 2018). For example, Gaier & Ha (2019) described an elegant architecture search by de-emphasizing the importance of weights. By utilizing a shared weight parameter they were able to develop ever-growing networks that acquired skills based on their interactions with the environment. However, given the brain's excitatory and inhibitory connections there is a rigidity to the weights that biological neural networks actually use.

Despite the weight implication, the principle of minimizing parameter count that Gaier & Ha (2019) addressed is productive when conceiving of biologically inspired artificial neural networks. In practice, neural networks tend to be over-parameterized, making them highly energy and memory inefficient. There has been a lot of work in the machine learning field of sparsity and low precision weights to alleviate these prominent issues. Sparsity of networks can be introduced prior to training, as shown by SqueezeNet (Iandola et al., 2016) and MobileNet (Howard et al., 2017). These networks were carefully engineered to have an order of magnitude fewer parameters than standard architectures while performing image recognition. Sparsity can also be introduced while training, as shown by Louizos et al. (2017); Srinivas & Babu (2015) where they explicitly prune and sparsify networks during training as dropout probabilities for some weights reach 1. Additionally, sparsity can be added after training is complete.

In this paper, we will leverage prior work in neuroscience, architecure search, sparse networks, and binary networks to demonstrate the presence of architecture agnostic neural networks, architecture agnostic neural network families, and the stochastic search and succeed algorithm's ability to navigate through the architecture manifold.

## 3 ARCHITECTURE AGNOSTIC NEURAL NETWORKS FORMULATION

### 3.1 SPARSE BINARIZED NEURAL NETWORKS

**Preliminaries** We represent a feed-forward neural network as a function $f(\mathbf{x}, \mathbf{w})$, that maps an input vector, $\mathbf{x} \in \mathbb{R}^k$, to an output vector, $f(\mathbf{x}, \mathbf{w}) = \mathbf{y} \in \mathbb{R}^m$. The function, $f(\mathbf{x}, \mathbf{w})$, is parameterized by a vector of weights, $\mathbf{w} \in \mathbb{R}^n$, that are typically set in training to solve a specific task. We refer to $W = \mathbb{R}^n$ as the *weight space* $(W)$ of the network. Here, $k$ is the input dimension, $m$ is the output dimension and $n$ is the total number of parameters in the neural network.

In this paper, we use two different neural network architectures for the static and dynamic task respectively. For the static task (e.g., MNIST classification): the network has 2 convolutional layers (16 filters 5 x 5) , 2 max-pooling layers and 1 fully-connected layer (1568 x 10). For the dynamic task (e.g., imitation learning for car-racing): this network has 2 convolutional layers (32 filters 7 x 7, 64 filers 5 x 5), 2 max-pooling layers and 2 fully-connected layers (576 x 100, 100 x 3).

**Sparse Binarized Neural Network** Throughout this paper, a binarized neural network refers to networks with weights constrained to (-1, 0, +1). We also constrain the output from every neuronal unit in the network to be in the range [-1, +1] by applying a binarized activation function. We use a "HardTanh" function defined as follows:

$$HardTanh(x) = \begin{cases} +1 & \text{x > 1} \\ -1 & \text{x < -1} \\ x & \text{otherwise} \end{cases} .$$

A p-sparse binary network $(w_b)$, which is a network with $p$ percent sparsity, is defined as follows:

$$||w_b||_0 = \frac{n*p}{100}; \text{ where, } w_b \in [-1,0,1]^n.$$

A $p$-sparse network refers to a neural network that has $\frac{n*p}{100}$ weights out of the total $n$ weights in the network set to 0.

## 3.2 Architecture Manifold and Network Families

As the weights of the binarized neural network are restricted to (-1, 0, +1), the architecture manifold follows the same definition as the weights space ($W$), defined above. Each point in the n-dimensional space, corresponds to a network with a distinct architecture.

We define nearest-neighbor architectures as networks that can be obtained by a single bit-swap of a non-zero connection (-1 or +1 weight) with a connection that didn't exist earlier (0-weight). For instance, if a network ($N_0$) is parameterized by $w_o$ = [1,0,-1,0,1,-1], some of its neighbors are as follows: $N_1$: $w_1$ = [0,1,-1,0,1,-1], $N_2$: $w_2$ = [1,0,-1,1,0,-1], $N_3$: $w_3$ = [1,0,-1,-1,1,0], $N_4$: $w_4$ = [1,-1,0,0,1,-1]. The original network $N_0$ and its neighbors ($N_1$ to $N_4$) are visualized in Figure 1.

A family of networks refers to a group of binarized networks at the same level of sparsity with the same number of (-1, 0, +1) weights in all layers.

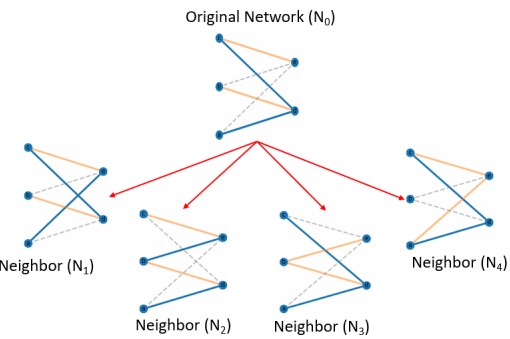

Figure 1: First-neighbors of a network ($N_0$) obtained by a bit-swap between a non-zero weight and a zero weight. The solid orange lines denotes a -1 weight, the dashed gray line represents a 0-weight connection (no connection), and the solid blue lines denotes +1 weight.

.

## 4 Learning Rules

Inspired by biological neural networks, we develop the SENSE algorithm to discover architecture agnostic artificial neural networks. As sparse binarized networks capture many salient properties of living networks, we begin by proposing a viable strategy to generate **p**-percent **s**parse b**i**na**r**y n**e**ural **n**etworks (p-SIREN algorithm), followed by the stochastic search and succeed (SENSE) algorithm to discover families of networks that maintain the same sparsity while being architecturally distinct.

### 4.1 p-SIREN: Generating Sparse Binary Neural Networks

The protocol for generating high-performance, $p$-sparse binary networks for the static MNIST task and the dynamic car-racing imitation learning task are as follows:

---

**Algorithm 1: p-SIREN:** Generate a p-sparse binary neural network.

---

**Result:** A high-performing p-sparse binary neural network.

**Step-1**: Initialize and train a dense neural network.

- Initialize dense neural net (**A**), s.t $A(\mathbf{w}) \in \mathbb{R}^N$, and **w** are real-valued.

- Train: backprop(A) = T for $n_1$ epochs.

- Final network: **T** has high performance and $T(\mathbf{w}) \in \mathbb{R}^N$

**Step-2:** Gradual "sparsification" of lowest magnitude weights to obtain p-sparse networks. Sparse networks denoted by $S_x$.

- $T(\mathbf{w}) \longrightarrow S_1(\mathbf{w}) \longrightarrow S_2(\mathbf{w}) ... \longrightarrow S_{n_2}(\mathbf{w})$, wherein $||T(w)||_0 = N$, $||S_1(w)||_0 = \frac{N*p}{100n_2}$ and $||S_k(w)||_0 = \frac{N*p*k}{100n_2}$

- $T(\mathbf{w}) \in R^N \longrightarrow S_{n_2}(\mathbf{w})$ after $n_2$ epochs, where $||T(w)||_0 = N$ and $||S_{n_2}(\mathbf{w})||_0 = \frac{N*p}{100}$.

**Step-3:** Backprop $(S_{n_2}(\mathbf{w})) \longrightarrow S_{n_3}(\mathbf{w})$ after $n_3$ epochs, wherein $||S_{n_2}(\mathbf{w})||_0 = ||S_{n_3}(\mathbf{w})||_0 = \frac{N*p}{100}$

**Step-4:** Binarize p-sparse neural network.

Binarize $(S_{n_3}(\mathbf{w})) \longrightarrow B(\mathbf{w})$. The networks' weights ($B(\mathbf{w})$) are clamped to $\{-1, 0, 1\}$. backprop $(B(\mathbf{w})) \longrightarrow B_{n_4}(\mathbf{w})$ after $n_4$ epochs.

---

This training procedure results in high-performance sparse binary neural networks for the tasks of interest. Figure 2 demonstrates the test accuracy of $n = 3$ binary neural networks at variable sparsities for both MNIST and car-racing imitation learning. We evaluate Figure 2a at 15 different sparsity levels (0%, 10%, 20%, 30%, ..., 80%, 85%, 87%, 89%, ..., 95%) on an MNIST test dataset. Figure 2b evaluates the dynamic car imitation learning test accuracy on the same sparsity levels as Figure 2a as well as (96%, 97%, 98%, 99%) for a total of 19 different sparsity levels.

We have attached a video of our 90% sparse binary network perform dynamic imitation learning on a car-racing task in the openAIgym environment. Find it in the supplement.

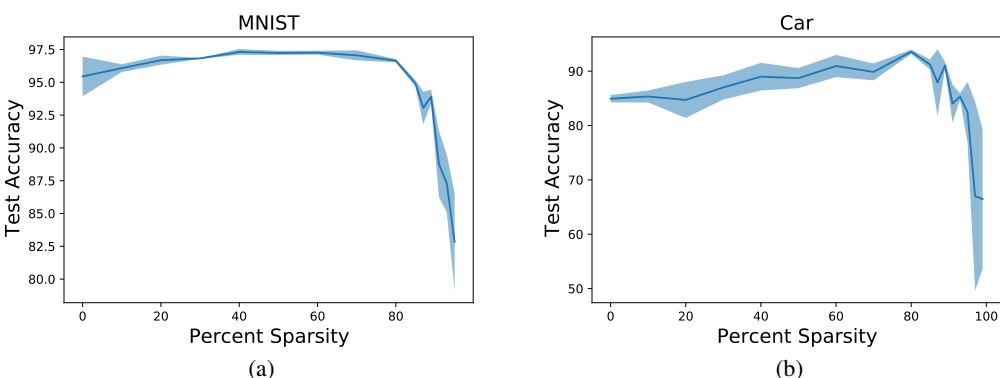

Figure 2: High-performance binary networks of variable sparsities. (a) A static task, MNIST classification. (b) A dynamic task, Car-racing imitation learning.

## 4.2 STOCHASTIC SEARCH...

After obtaining a high-performing sparse binary neural network that works on either a static or dynamic task, we explore its local neighborhood on the architecture manifold using random bit-swaps. A single random bit-swap within the network generates architecturally distinct first-neighbors. Two random bit-swaps generates second-neighbors, so on and so forth. The bit-swap procedure to generate first neighbors of a sparse binary network is explained in detail in section 3.2 (Architecture Manifold and Network Families).

**Finding local neighbors:** On applying a single bit-swap to the original performant network multiple distinct times, we generate a large number of first neighbors that maintain the same level of sparsity.

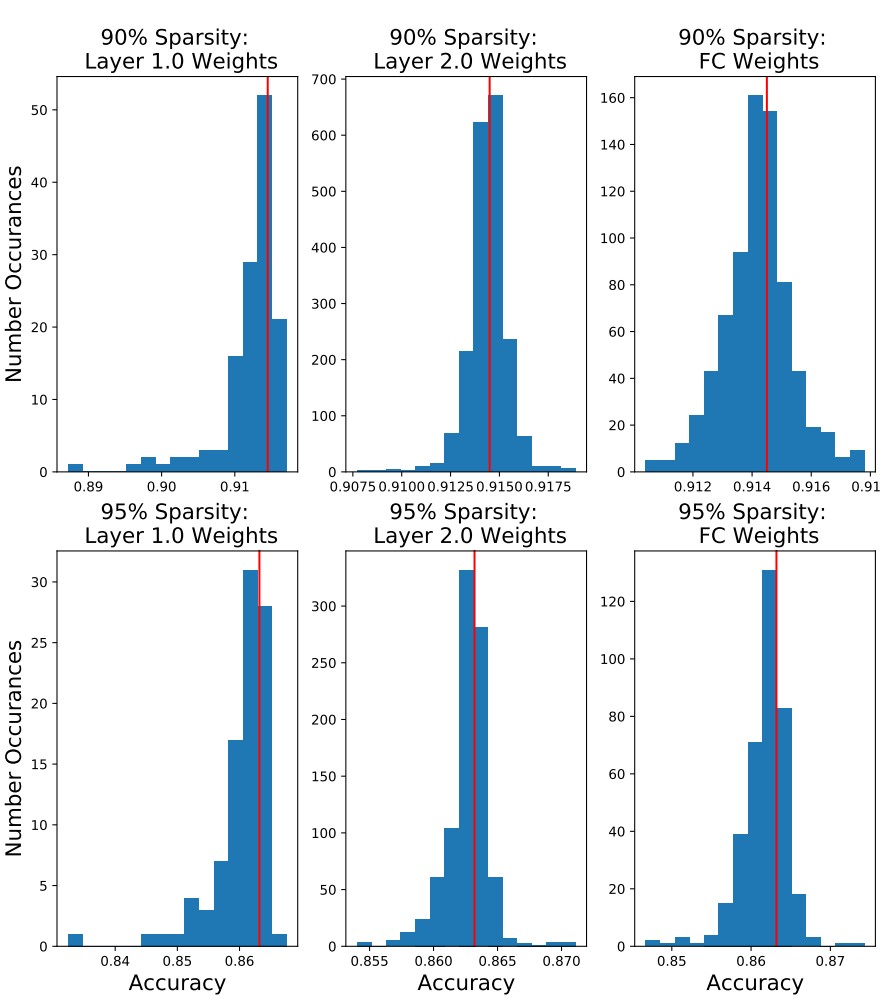

Figure 3: MNIST task test accuracy distribution of immediate family neighbors. We observe that as we move into deeper layers a larger percentage of the first family nearest neighbors outperform the original model accuracy.

We perform bit-swaps in a layer dependent manner; convolutional layers or fully-connected layers can be perturbed. We then evaluate the test accuracy of the first neighbors and plot a histogram of their performance, relative to the performance of the original network (depicted by a vertical red line) in Figure 3. The same procedure was applied to sparse binary networks trained to perform the car-racing imitation learning task, shown in Figure 9.

We observe that perturbation of sparse binary networks in the higher layers (2.0, FC) gives rise to a large number of sparse binary networks that perform better than the original network, while the lower layers (1.0) seem to be more tightly optimized: Fig 3, Fig 9. It is interesting that we discover more generalizability at higher layers, especially since within the brain, sensory neurons lower in the network hierarchy are more tightly optimized than cortical structures (Kara et al., 2000).

The variable robustness to bit-swaps across multiple layers in the sparse binary network begs the question: what is the nature of sparse binary networks' loss manifold?

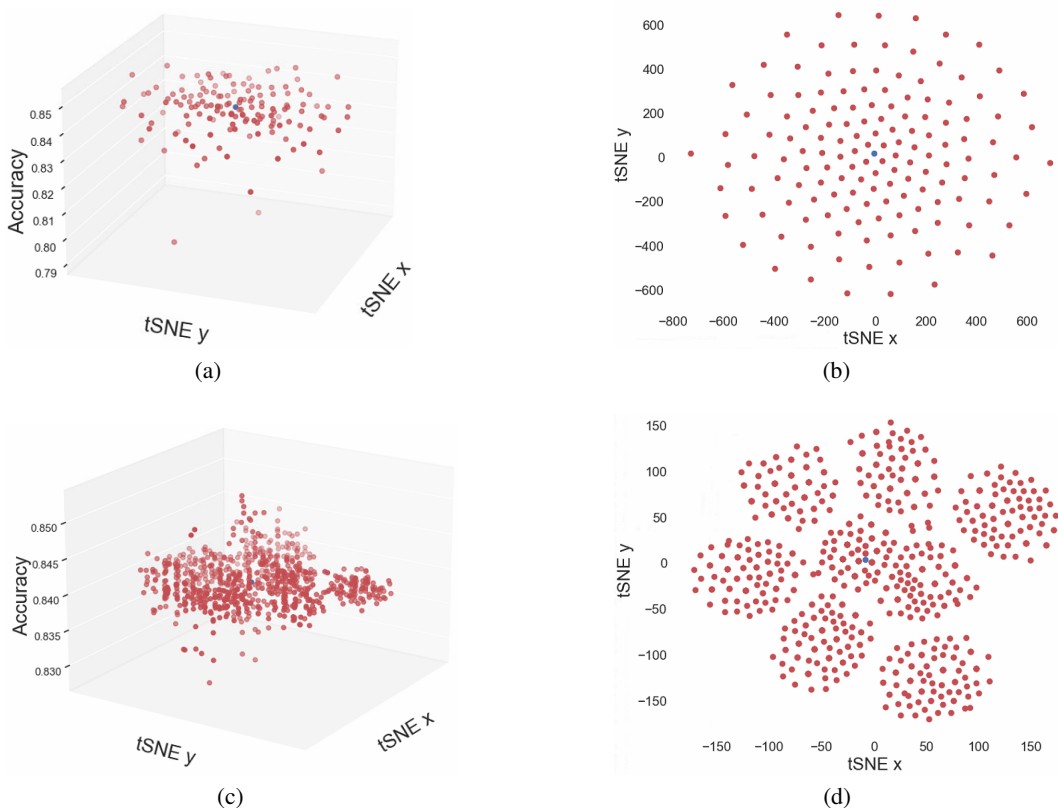

Figure 4: First neighborhood families for 90% sparse binary network trained on MNIST projected via tSNE accompanied by network test accuracies. Blue dot represents the original network. (a) Side view of first neighbors from layer 1.0 weight perturbations. (b) Top view of first neighbors from layer 1.0 weight perturbations. (c) Side view of first neighbors from fc weight perturbations. (b) Top view of first neighbors from fc weight perturbations. We see that as we move into deeper layers the first family nearest neighbors cluster more than the first family nearest neighbors in the initial layers.

In order to study the loss manifold of sparse binary networks, we visualize the first-neighbors of the generated high-performing sparse binary network using tSNE, a tool that enables non-linear dimensionality reduction of the high-dimensional manifold onto 2D and 3D space.

In Figure 4, the x,y axes represents tSNE axis, while the z-axes represents the test accuracy on the MNIST dataset. We notice multiple local clusters of first-neighbors particularly in Fig 4d, possibly indicating symmetry of first neighbor networks obtained after performing the stochastic weight-swaps. In addition, each local cluster of networks has a large range of performance, as seen in Fig 4c, but more predominantly in Fig 10. The presence of local clusters in the architecture manifold indicates that the stochastic search and succeed algorithm can enable a quick survey of a larger "area" of the architecture manifold at each step and wouldn't be constrained to a small epsilon-neighborhood that the conventional backpropagation algorithm faces.

## 4.3    ... AND SUCCEED

As the stochastic search for first-neighbors results in a large proportion of better performing networks for the task of interest, we implement a stochastic search and succeed algorithm to survey a small set of local sparse binary networks, pick the one that performs the best, and obtain the first neighbors for the best first-neighbor network chosen. This step can be repeated multiple times to find binary networks of the same sparsity perform much better than the original sparse binary neural network obtained! The results for a 90% MNIST network climbing ~3% in accuracy over 6 neighborhoods can be seen in Fig 5. This indicates that SENSE increases the accuracy of the model even after the model has been trained to saturation (to a local optimum) using back-propagation! It also indicates that

searching and optimizing in the architecture manifold is an effective way to improve the accuracies of the model.

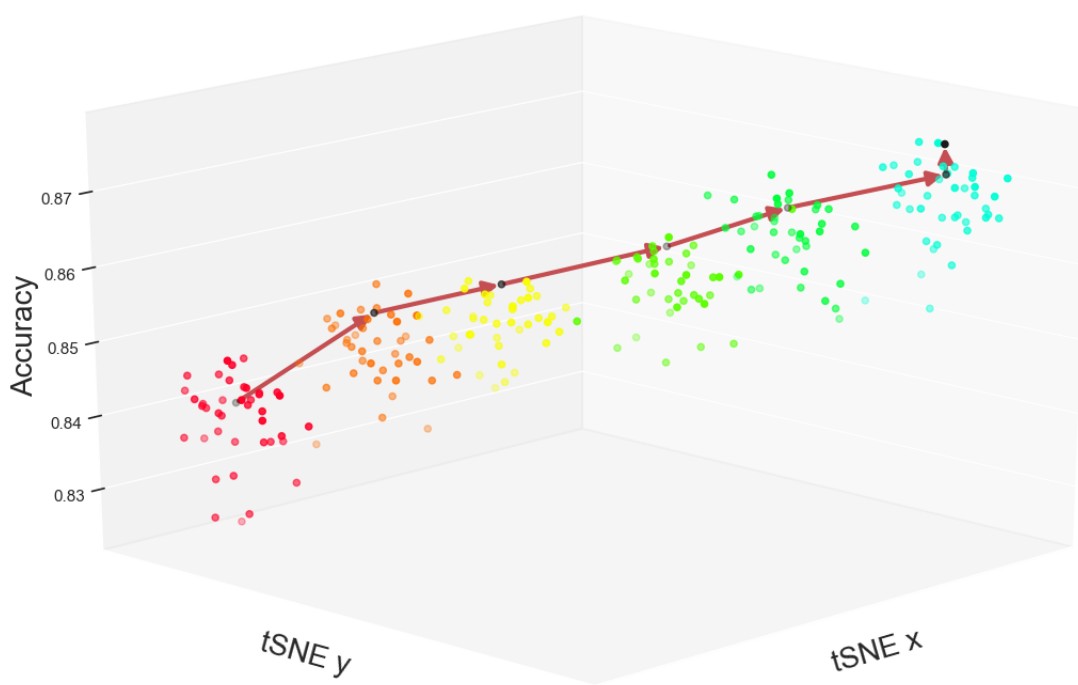

Figure 5: 90% MNIST network climbing 3% in accuracy over 6 neighborhoods. Each neighborhood has a different color. Black dots represent the best neighbor in that family. Red lines indicate the path from best neighbor to best neighbor that the algorithm takes. We see that neighborhoods cluster with one another (i.e. first nearest neighbors cluster, second nearest neighbors cluster, etc.). In addition, we see how neighborhoods build on one another to move in the architecture manifold.

---

**Algorithm 2: SENSE:** Stochastic Search and Succeed

---

**Result:** Family of high-performance sparse binary neural networks.

**Initialization**:
  Without Backprop: Begin with a randomly initialized neural network ($N^0$).
  **OR**
  With Backprop: Begin with a p-sparse binary neural network ($N^0$) from Algorithm 1 p-SIREN (p-SIREN utilizes backprop to create its model).

**Then** Set number of nearest neighbors = $n_k$; evaluate accuracy of network $N^0$ ($A(N^0)$); Set max-acc = ($A(N^0)$); Set ii = 0;

**while** $ii < n_k$ **do**
    Find all networks in the local neighborhood of $N^{ii}$ through random bit-swaps within the network. [$N_1^{ii+1}$, $N_2^{ii+1}$, ...]; Evaluate test accuracy of all local neighborhood networks. [$A(N_1^{ii+1})$, $A(N_2^{ii+1})$, ... ]
    **if** $max\text{-}acc > A(N_j^{ii+1})$ **then**
        No better performing neighbor network. Stop current search; pick another layer to perturb or retry the same layer with a different random initialization.
    **else**
        Pick a neighbor network ($N_j^{ii+1}$) with accuracy > max-acc; set max-acc = $A(N_j^{ii+1})$
    **end**
**end**

---

## 4.4 RANDOM INITIALIZATION

We first train a 90% random sparse binary neural network on MNIST through the stochastic search and succeed (SENSE) algorithm. In Figure 6, we alternate the stochastic search throughout layer 1.0 (red) , 2.0 (green) and fc (blue) for 10 epochs each and then repeat the permutations 3 times. We can see that by the third epoch the 1.0 and 2.0 layers plateau while the fc layer continues rise with only a moderate plateau, perhaps attributed to the fact that fc is the deepest layer. In both figure 6 and figure 7 the colored line plots the validation accuracy with the SENSE algorithm The black line with orange markers denote the test accuracy of the SENSE algorithm. The black line with violet markers denote the test accuracy of utilizing backpropagation on the initial sparse, binary network.

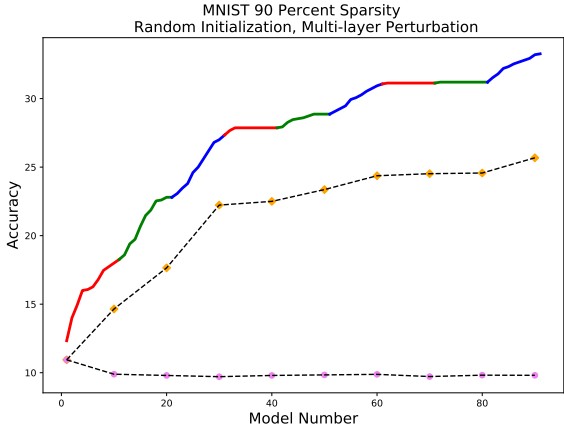

Figure 6: The filled in and colored line denotes the validation accuracy; red indicates bit switches in layer 1.0, green denotes bit switches in layer 2.0, and the blue denotes bit switches in the fc layer. Colored line begins at a validation accuracy of 12.34, and a final validation accuracy of 33.27. Black and orange begins as a test accuracy of 10.94, and reaches a final test accuracy of 25.68. Black and violet begins as a test accuracy of 10.94, and reaches a final test accuracy of 9.81.

Given the fc layer's success, we trained another model with the SENSE algorithm where we only permuted the fc layer weights. Results for this experiment are shown in figure 7.

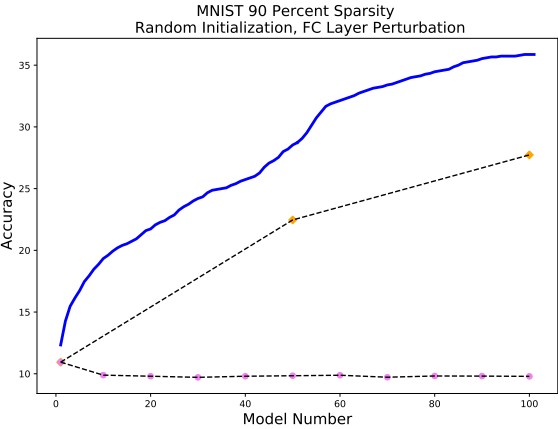

Figure 7: The filled in and colored line denotes the validation accuracy; red indicates bit switches in layer 1.0, green denotes bit switches in layer 2.0, and the blue denotes bit switches in the fc layer. Colored line begins at a validation accuracy of 12.34 and reaches a final validation accuracy of 35.87. Black and orange begins at a test accuracy of 10.94 Reaches a final test accuracy of 27.73. Black and violet begins at a test accuracy of 10.94 Reaches a final test accuracy of 9.79.

We then wanted to explore how a randomly initialized network would perform if we randomly selected a layer to perturb. Utilizing 80% random sparse binary neural networks on both MNIST, figure 8, and on car racing, figure 8, we randomly select the layer (layer 1.0, layer 2.0, or the fc layer), and then permute the architecture for three successive neighborhoods. Once we have permuted the first randomly chosen layer for 3 neighborhoods, we randomly select another layer and repeat. With MNIST and car-racing we reach test accuracy values of 63.4% and 74.0% respectively without utilizing backpropagation.

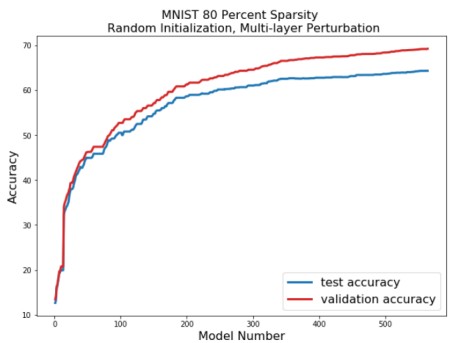

(a) The validation line begins at 13.45% and ends at 69.5%, test line begins at 12.6% and ends at 63.4%.

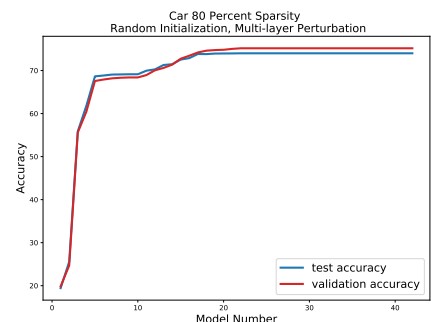

(b) The validation line begins at 19.8% and ends at 75.17%, test line begins at 19.0% and ends at 74.0%.

Figure 8: We start from a randomly initialized network, chose a layer (layer 1, layer 2, or fc layer in the MNIST case; layer 1 or layer 2 in the Car-racing case) at random, and then permute the layer with three successive bit-swaps spanning three neighborhoods. Finally, we repeat the last two steps many times to reveal networks solely trained by our SENSE algorithm.

## 5 CONCLUSION

In this paper, we create a stochastic search and succeed algorithm to show: (i) The first demonstration of artificial, architecture agnostic neural networks (AANNs), that retain biological plausibility. AAAN refers to a family of networks that are functionally similar but architecturally distinct. (ii) That network families with common architectural properties share similar accuracies and structural properties. (iii) That moving in the architecture manifold improves performance with both randomly initialized and backpropagation trained models. In the process we generate an original sparse binary network and then explore its neighbors through the architecture manifold. Our stochastic exploration is inspired by developmental stochastic pruning and biological network's ability to maintain high-performance on tasks performed while undergoing pruning.

In the future, we plan to survey the architecture manifold by appending principles from simulated annealing to our SENSE algorithm. In addition, we would like to assess the generalizability of sparse binary networks as well as its effectiveness during transfer learning. Finally, we believe that utilizing the network families in ensembles would lead to robust performance both within and across tasks.

By optimizing sparse binary neural networks architectures we will eventually be able to uncover more broad principles of neural network architectures and move the community away from hand-crafted architectures. In addition to uncovering neural principles, there is the additional advantage that sparse binary networks consume less power and utilize less memory making them a suitable model class to operate on edge devices.

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

# A APPENDIX

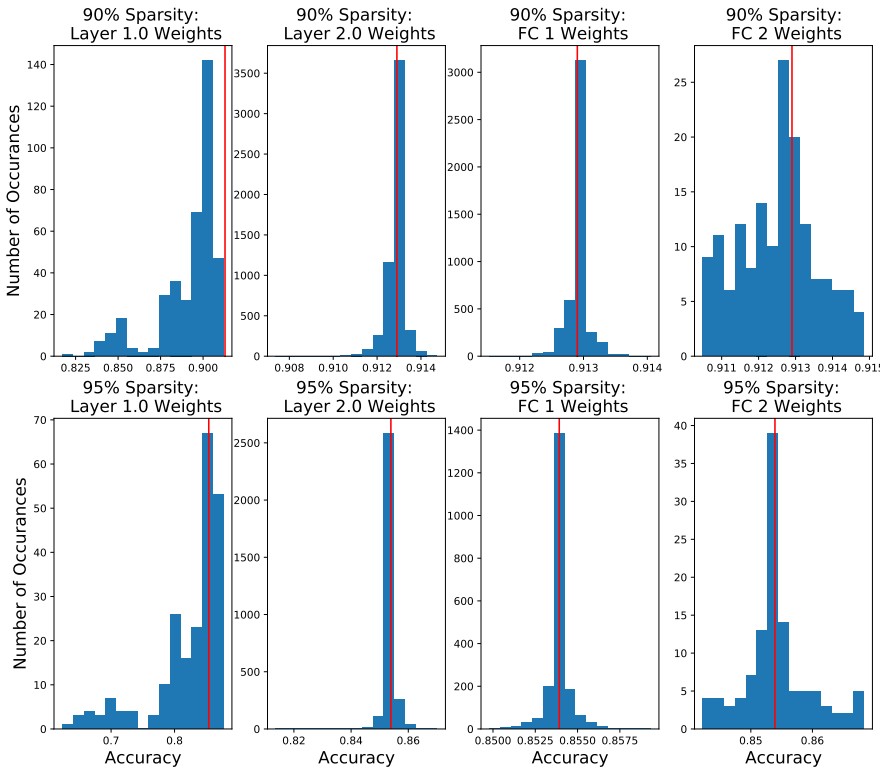

Figure 9: Car task test accuracy distribution of immediate family neighbors
.

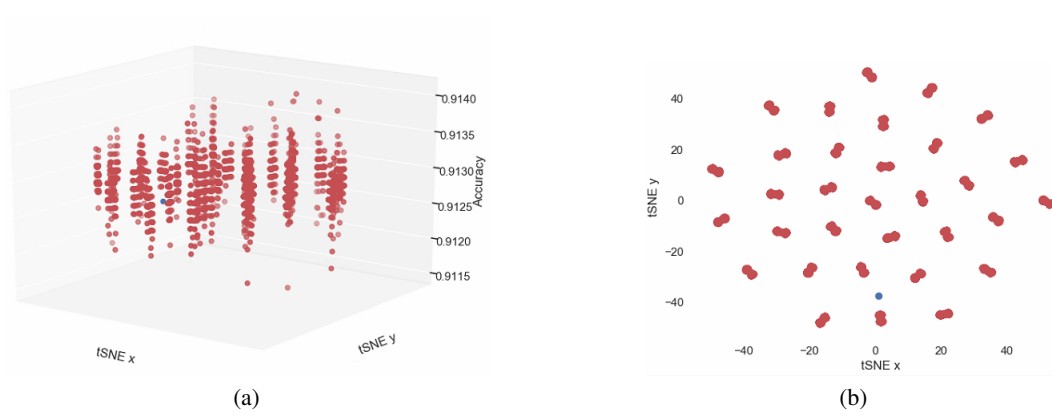

Figure 10: First neighborhood families for 90% sparse binary network trained on car-racing projected via tSNE accompanied by network test accuracies. Blue dot represents the original network. (a) Side view of first neighbors from fc layer perturbations. (b) Top view of first neighbors from fc layer perturbations.

.

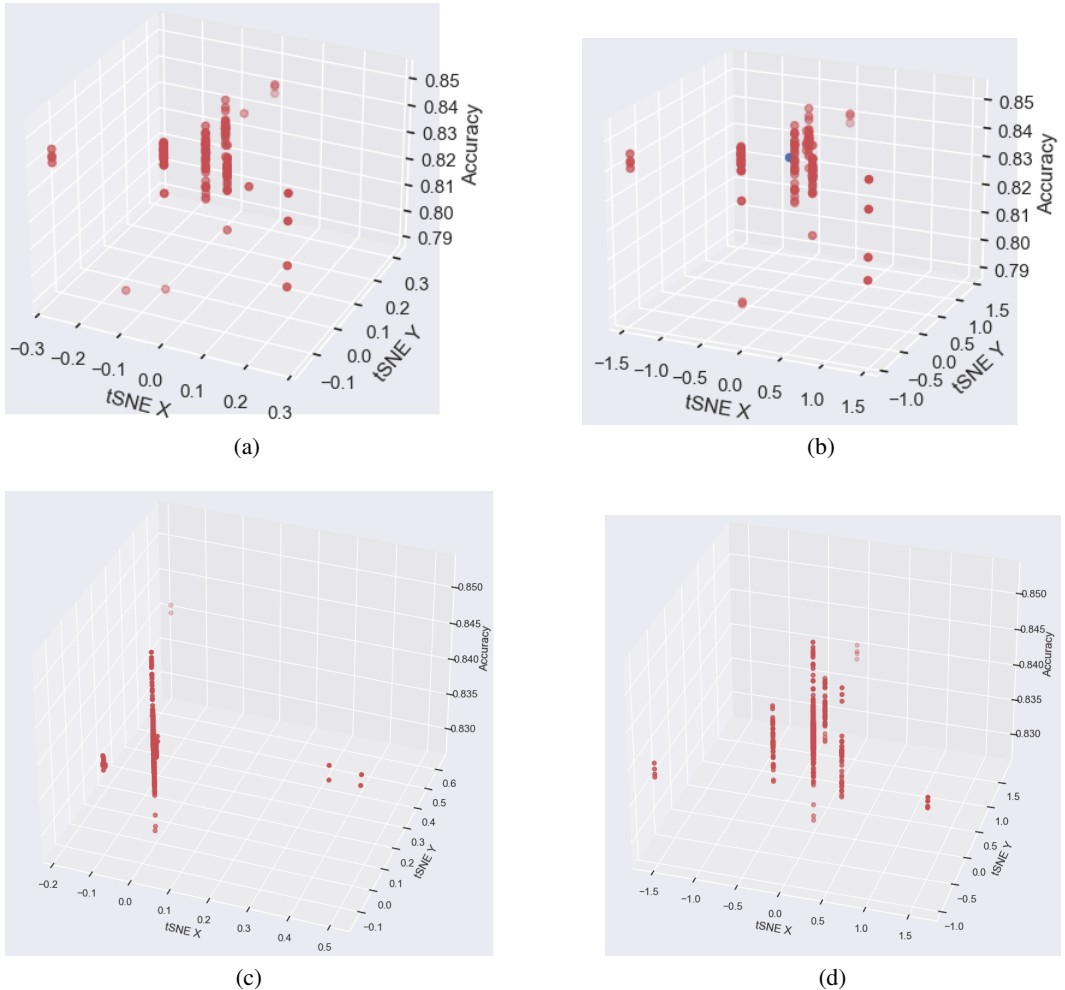

(a)  (b)

(c)  (d)

Figure 11: First neighborhood families from layer 1.0 perturbations for 90% sparse binary network trained on MNIST projected via (a) Locally linear embedding (LLE), (b) Isomap accompanied by network test accuracies. First neighborhood families from fc perturbations for 90% sparse binary network trained on MNIST projected via (c) LLE, (d) Isomap accompanied by network test accuracies.

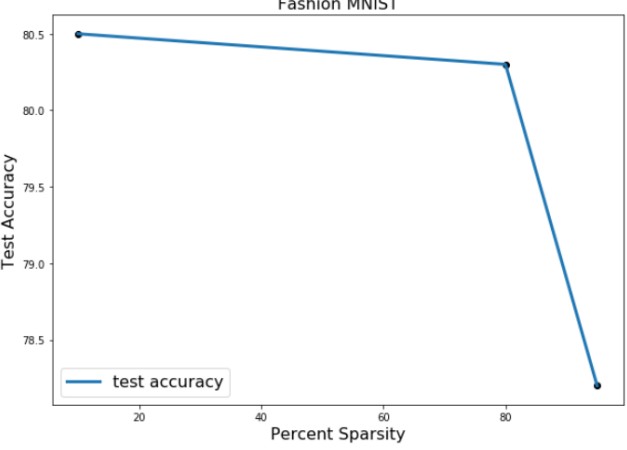

Figure 12: High-performance binary networks of variable sparsities on static task, Fashion MNIST
.

