# OpenReview forum: "Architecture Agnostic Neural Networks"
_ICLR.cc/2021/Conference — Reject_

### Official Review · AnonReviewer3 · 2020-10-28
**Official blind review 3**

**Rating:** 5
**Confidence:** 5

**Review:**

Summary:
In this paper, the authors have explored a "brain’s stochastic synaptic pruning" inspired method for architecture agnostic models.
Authors have explored sparse and binary paradigms for neural architecture and architecture manifold using random bit-swaps.
Authors have tested their methods on a static and dynamic tasks.

Strengths:
Both sparse, binary paradigms for neural architecture and sampling using random bit-swaps are priorly less explored tasks, with very less literature.
The accuracies listed in paper motivates us that the authors are taking right steps towards brain-like-architectures.

Weakness:
The datasets used are smaller and not diverse. The authors did not explain the reasons for architecture choice and epoch choice for sparse binary networks.
overall, the authors should take time to explain/correct the following things:
- Can restricting weights to binary format impact generalization of network? How good is the network during transfer learning?
- The authors coul have tried other small datasets. Why only MNIST?
- Why the parameters in the architecture are choosen the way ther are presented.
- In GENERATING SPARSE BINARY NEURAL NETWORKS section - Abalation study would have helped understand the choice of stages. What are the reasons for choosing 4 stages of epochs?
- Diagrams are bit unclear. Colour representations in tSNE are not clearly visible on the PDF.

---

> ### Author Response · Authors · 2020-11-25
> **Reviewer 3 Response**
>
> Reviewer 3, thank you for the time you took to read our submission, and for your feedback! We’re happy to hear that you think we are taking the right steps towards brain-like-architectures.
>
> Comments on your points:
>
>
> 1) Studying the generalizability of these networks is something that we are very interested in because the brain utilizes small architecture perturbations while robustly engaging in many scenarios and tasks. In addition to seeing how the networks perform during transfer learning, we want to see how we can leverage the network families to produce robust ensembles. We have updated our future directions to include these points.
>
>
> 2) In the original submission we used the first algorithm on both MNIST and the dynamic car-racing task, but only demonstrated the results from the second algorithm “stochastic search and succeed” on MNIST. In our updated version we demonstrate the stochastic search and succeed algorithm on a randomly initialized (no backprop) MNIST and car-racing networks. We achieved an MNIST test accuracy of 63.4% (started from 12.6%, this is also higher than reported in the first submission), and a car-racing test accuracy of 74.0% (started from 19.0%). These results can be seen in the new figure 8. We are also testing on Fashion MNIST and have preliminary results for the first algorithm, figure 12, and will continue to generate results with Fashion MNIST.
>
>
> 3) We believe the reviewer is asking about why we chose the specific number of layers, types of layers, and layer sizes. We wanted to understand how the concept of biologically plausible architecture agnostic neural networks would function in an in-silico context. Starting with smaller networks allowed us to quickly probe interesting similarities between our neural networks and the brain when it undergoes developmental synaptic pruning. We mainly wanted to study: if biologically plausible architecture agnostic neural networks existed, if network families shared similar accuracies and structural properties, if we could improve performance by moving in the architecture manifold. Using small networks were a tool that allowed us to study these properties rapidly.
>
>
> 4) We chose these four stages as well as their order via numerous bouts of ablation studies.  For instance, in step-2 we tried to delete all of the lowest magnitude weights at once. However, especially for the higher sparsity networks, the resulting network wouldn’t be able to learn no matter how much training we applied. Only with gradual deletion were we able to have high-functioning sparse binary neural networks. This was an interesting parallel for us, since the brain also doesn’t delete all of its synaptic connections in one go. It takes years for developmental synaptic pruning to occur.
>
>
> 5) We have updated our diagram for the color tSNE representations to be clearer.

---

### Official Review · AnonReviewer4 · 2020-10-28
**An interesting idea for random architecture search for binary sparse networks**

**Rating:** 4
**Confidence:** 4

**Review:**

The paper proposes a way to perturb a binary sparse NN in order to achieve a higher accuracy.

First they trained, binarized and sparsify a network with a couple of conv and FC layers. Then, they propose to create a swarm of networks by swapping a random non-zero weight with a zero weight.
Some of these networks happen to perform slightly better than the original one. Based on this the author propose a multi-stage algorithm that they call "a stochastic search and succeed algorithm" (SSS) that essentially alternates between training and swapping steps.

The paper also tries to analyze the manifold of the network weights of different swapped networks by visualizing them using t-SNE algorithm.

The evaluation is done in MNIST and car-racing dataset. Generally, I'm not convinced that the binary networks would perform much better beyond MNIST and car examples.

The paper raises a series of questions:
- How accurately was the network trained? Did the authors rained the best possible binary sparse MNIST network before preceding to the swapping. This is not clear from the text. It would be great to at least repeat the procedure they describe for training multiple times.
- How is the affinity matrix of tSNE is computed? I assume that if the input is binary, simple Euclidean distance won't be a great measure. Symmetric nature of the plots at Fig4 suggest that your perplexity parameter is too high. The results on the plots are rather the artifacts of the wrong affinity matrix than the properly of the weight manifold.
- The description of SSS lacks rigor. How many networks are selected in the first step? How do first neighbor are chosen? It is an exhaustive search? What is the complexity? What is the stopping criteria? What "stochastic" about this algorithm?

Nits:
- Blue marker is almost impossible to see on fig.5

---

> ### Author Response · Authors · 2020-11-25
> **Reviewer 4 Response**
>
> Reviewer 4, thank you for your feedback, and the time you took to read our submission!
>
> Before we dive into your specific questions, we’ll take some time to discuss some of the points in your summary.
>
> You state that “Some of these networks happen to perform slightly better than the original one” after the stochastic weight swaps. When permuting in later layers, roughly 50% of networks perform better than the original in both our static and dynamic tasks (as evidenced by figures 3 and 8 in the original submission), this qualifies as more than “some” networks. Moreover, although the accuracy does “slightly” increase with a single swap, we have demonstrated that by building upon swaps we can move through neighborhoods to reach dramatic accuracy increases, as shown in figure 5, figure 6, figure 7. In addition, in our updated paper we demonstrate that on a randomly initialized (no backprop) MNIST and car-racing networks the “stochastic search and succeed” algorithm achieves higher test accuracies than previously reported. We achieved an MNIST test accuracy of 64.34% (started from 12.6%) and a car-racing test accuracy of 74.0% (started from 19.0%).
>
> You state that our stochastic search and succeed algorithm “essentially alternates between training and swapping steps.” There is explicitly no training in this algorithm. We only permute networks and then test the new network to see the resulting accuracy. Training would defeat the purpose of the algorithm.
>
> We are sorry to hear that you’re “not convinced that the binary networks would perform much better beyond MNIST and car examples.” We intentionally chose two tasks that were drastically different from one another to demonstrate robustness. Since MNIST classification is a static task, while dynamic imitation car-racing is a dynamic task we feel we’ve hit the mark of sufficient difference. However, we have started to run the tasks on Fashion MNIST and have preliminary results on the first set of experiments in figure 12 of the updated paper.
>
> 1. In the case of the first algorithm, where we use backpropagation to setup the network, we train the networks with n1, n2, n3, n4 epochs. We heuristically set the number of epochs depending on the desired network sparsity, such that the network’s accuracy will plateau by the end of n4 epochs. We have included a more mathematical algorithm block in the new submission that is hopefully more clear.
>
> For the first algorithm we do repeat the experiments (3x each) on multiple sparsity levels for both MNIST, and dynamic car-racing in figure 2. That is how we were able to derive the error bounds seen in the figure.
>
> 2. To demonstrate that our tSNE representation isn’t an artifact, in our new version we visualize the architecture manifold using Isomap and locally linear embedding (LLE) that use the geodesic distance on the manifold to as their distance metric, instead of using euclidean distances as done in tSNE (figure 11). You can see that the networks cluster in the columnar groupings that we previously observed.
>
> 3. The number of nearest neighbors (nk), see algorithm block 2, is the number of neighborhoods you will search through, see figure 5. Our nk-s correspond to the x-axis in figure 6 and figure 7 in the original submission and figure 6, figure 7, and figure 8 in the new version.
>
> Once the user decides how many neighborhoods they want to search through (nk), in each neighborhood every valued weight (-1, +1) is swapped with a randomly chosen 0-weight. Since the 0 weight is randomly chosen it is a stochastic choice. Moreover, the search is not exhaustive. In order to do an exhaustive search every valued weight would have to be swapped with every 0 weight. Each of these weight swaps would lead to a new network that would have to be tested. To contextualize: if you had 30 valued weights in a network and were performing the first set of weight swaps, there would be 30 stochastically derived first nearest neighbors.
>
> The stopping criteria depends upon nk. For instance, if I wanted to only search out to the 5th neighborhood and my original network had 30 valued weights. I would have 30 first nearest neighbors, 30 2nd nearest neighbors, … , 30 5th nearest neighbors and then the algorithm would stop.
>
> We have updated the algorithm block in the new submission to clarify what we had previously written.
>
> 4. We have changed figure 5, such that both neighborhoods and best networks in neighbors are easier to see.

---

### Official Review · AnonReviewer1 · 2020-10-29
**Interesting work**

**Rating:** 5
**Confidence:** 4

**Review:**

The authors pay attention to Architecture Agnostic Neural Networks. I think their stochastic search algorithm is a kind of EA method. Start with a initial architecture and then swap a single bit-swap to obtain the child architecture. The pruning method in learning rule is the standard magnitude-based pruning. Thus, from the view of technique, the contribution is somewhat weak, even though the conclusion of this paper is interesting.

I am confused about the title of section 4.2 and section 4.3. Is it incomplete?  Moreover, I think stochastic search is not suitable
 for your algorithm since the whole procedure is much similar with evolutionary algorithm. The only difference is how to define mutation for Architecture Agnostic Neural Networks.

---

> ### Author Response · Authors · 2020-11-25
> **Reviewer 1 Response**
>
> Reviewer 1, thank you for the time you took to read our submission, for your feedback, and for believing that the conclusion of the paper is interesting!
>
> On some of your points:
>
> We will be utilizing the definition of evolutionary algorithms (EAs) found at the bottom of this review for our response.
>
> The reviewer points out that our approach is a variant of the EA algorithm. EA methods encompass a broad array of strategies, since the population, fitness value, and variation can differ greatly, so the unique details are important. Here, we’d like to highlight that most EA based strategies aren’t biologically plausible. In addition, as discussed in our global comment, our algorithm truly is a tool to study the neuroscientific principles that have largely eluded the machine learning community. We believe that these principles discovered via “stochastic search and succeed”:
>
>         - The first demonstration of artificial, architecture agnostic neural networks, that retain biological plausibility.
>         - The first demonstration of artificial, architecture agnostic neural networks, that retain biological plausibility.
>         - The demonstration that network families with common architectural properties share similar accuracies and structural properties.
>
> show the importance of architecture agnostic neural networks. We believe that this contribution to the intersection of neuroscience and machine learning is not weak, and in fact demonstrates very profound shared principles between what is possible in vivo and in silico.
>
> The titles in section 4.2 and section 4.3 are not incomplete. We broke up section 4.2 (Stochastic Search…) and 4.3 (... And Succeed) in this manner to walk the reader through the concepts without jumping into the dense math that generally accompanies an algorithm block. However, we have updated the section in the new version with a single algorithm block which hopefully clears up any confusion.
>
> In addition, we’d like to alert the reviewer to the fact that in our updated version we demonstrate that on a randomly initialized (no backprop) MNIST and car-racing networks the “stochastic search and succeed” algorithm achieves higher test accuracies than previously reported. We achieved an MNIST test accuracy of 64.34% (started from 12.6%) and a car-racing test accuracy of 74.0% (started from 19.0%).
>
> ------------------------------------------------------------------------------------------------------------------------------------
>
> Definition of Evolutionary Algorithms:
>
> “EAs are algorithms that perform optimization or learning tasks with the ability to evolve. They have three main characteristics:
>
> - Population-based. EAs maintain a group of solutions, called a population, to optimize or learn the problem in a parallel way. The population is a basic principle of the evolutionary process.
>
> - Fitness-oriented. Every solution in a population is called an individual. Every individual has its gene representation, called its code, and performance evaluation, called its fitness value. EAs prefer fitter individuals, which is the foundation of the optimization and convergence of the algorithms.
>
> - Variation-driven. Individuals will undergo a number of variation operations to mimic genetic gene changes, which is fundamental to searching the solution space.” [a]
>
> [a] Introduction to Evolutionary Algorithms https://books.google.com/books?hl=en&lr=&id=rHQf_2Dx2ucC&oi=fnd&pg=PR8&dq=evolutionary+algorithms&ots=xNxd1rwhCH&sig=YsQ0IKU9HO0NwHd2B_YFg0QE4F8#v=onepage&q=evolutionary%20algorithms&f=false

---

### Official Review · AnonReviewer2 · 2020-11-02
**Pruning and binarizing neural networks**

**Rating:** 4
**Confidence:** 4

**Review:**

In this paper, the authors study a procedure for pruning (sparsifying) and binarizing neural networks through a pruning procedure. They do this by taking a trained dense network, pruning the synapses to get to a sparser network, and then doing a stochastic search over "connection swaps" to further optimize the pruned network.

Reasonable performance is shown on MNIST. They also show data for a car-racing imitation task; the details of that task are a bit sparse, so I am not sure how impressive their 90% performance figure is for that task.

I found some other details to be missing (discussed below), and also have a few conceptual criticisms of this work. I like the concept a lot: of searching over sparse network configurations to find high performance small networks. But this work seems somewhat preliminary.

Criticisms:

1) Sec 4.1 could have used more detail:
a) how do you decide which connections to prune in step 2? Is it the weakest ones? Or did you find those for which the gradients of the loss with respect to the weights were smallest in magnitude? Or do something else?

b) what is the training procedure during step 4 (training after binarizing)? Is that a combinatoric search over the connection swaps? Or was this just done by adjusting the thresholds for individual units? Or some other thing?

2) Sec. 4.2: is the search over swaps greedy (one connection swap at a time)? If so, that seems likely to miss global optima that require, say, a "bad" bit swap to get over to a better region of the space. That should be discussed I think, even if there is not an immediately effective solution available.

3) This work doesn't seem architecture agnostic: you are still specifying the number of layers, conv vs dense, etc. It seems more like you have an approach for sparsifying (which could still be useful!). But I am not persuaded that this work solves the architecture search problems in any meaningful way. There has been some nice recent progress in this area (e.g., the autoML zero work from Quoc Le et al.) that might interest the authors if they are curious about genuine progress in architecture agnostic NNs.



- autoML zero
- doesn't seem architecture agnostic

---

> ### Author Response · Authors · 2020-11-25
> **Reviewer 2 Response**
>
> Reviewer 2, before we dive into addressing your criticisms we’d like to thank you for your time, your feedback, and for seeing the promise in our concept!
>
> Onto your criticisms:
>
> 1a) In the procedure: section 4.1, step 2, we mention that we use a magnitude-based mask to prune away the network’s synaptic weights. Canonically, magnitude-based masks indicate that the weakest weights are deleted; for instance in “Pruning Filters for Efficient ConvNets” [a] in the abstract they state that they use “magnitude-based pruning” and later on in the paper clarify that this means targeting “weights with small magnitudes”. In addition, we went on to state that “we then apply a magnitude-based mask at the end of every epoch to prune away (p/n2) % of the networks synaptic weights at the end of every epoch.” However, we can absolutely be more explicit and state that magnitude-based masks are used to prune away the weakest/smallest/lowest weights. In our new uploaded version, we have updated the explanation to be more mathematical (hopefully more clear!) and have included the language in the Algorithm 1 block:  “Step-2: Gradual “sparsification” of lowest magnitude weights to obtain p-sparse networks”.
>
> [a] Pruning Filters for Efficient CONVNETS https://openreview.net/pdf?id=rJqFGTslg
>
> 1b) For the first algorithm discussed in the paper:
> In step 1: we instantiate a real-valued dense network.
> In step 2: we sparsify this network over n2 epochs setting smallest weights to 0 with a magnitude-based mask.
> In step 3: we train the sparsified network for n3 epochs.
> In step 4: we binarize all the weights (which are valued) to be -1 or +1 by applying a hard sigmoid function.
>
> In the first algorithm, there is no combinatoric search over the connection swaps, or by adjusting the threshold for individual units. In our second algorithm, “stochastic search and succeed” we do stochastically perform weight swaps, which is perhaps what you are referring to. However, this does not occur in algorithm 1.
>
> We made algorithm 1 more mathematically precise our new uploaded version, so hopefully it is more clear now.
>
>
> 2) Yes, algorithm 2 “stochastic search and succeed” is greedy, and could miss the true global optima due to the selection of swaps that result in each neighborhood’s best network. However, we can easily address this by adopting methods such as simulated annealing. We have listed this as a future direction for our work.
>
>
> 3) We define architecture agnostic neural networks similarly to how weight agnostic neural networks [b] are defined: “neural network architectures that can already perform a task without any explicit weight training”. In our case, instead of changing the weights and expecting high-performance without retraining we keep the weights fixed and change the architecture while expecting high-performance without retraining. Both of these definitions build off of extensive work in neuroscience to ground our respective studies with biological meaning. We are not interested in high-performance or robustness if it is devoid of insight.
>
> We define a neural architecture by:
> - the number neurons
> - the allowed (i.e. non - 0) weights between neurons
>
> Therefore, by changing either the number of neurons or the weights between neurons we are changing the neural architecture. Furthermore, by stochastically forcing the weights of wildly different connections to be 0 while maintaining performance, we believe our networks fit the broader definition of architecture agnostic neural networks. This definition is consistent with the brain’s changing architecture during developmental synaptic pruning, as elaborated on in the paper. These types of studies can be interesting for many reasons, including 1) identifying the possible architecture, 2)  discovering “machine learning algorithms from scratch” [c], and 3) better understanding biological processes.  While the bulk of neural architecture search is focused on the first two points, we chose to investigate the latter.
>
> We therefore disagree that working to utilize a more biologically plausible system makes our definition of architecture agnostic neural networks lack “genuine progress” or that we aren’t approaching the problem in a “meaningful way”.
>
>
> 4) In addition, we’d like to alert the reviewer to the fact that in our updated version we demonstrate that on a randomly initialized (no backprop) MNIST and car-racing networks the “stochastic search and succeed” algorithm achieves higher test accuracies than previously reported. We achieved an MNIST test accuracy of 64.34% (started from 12.6%) and a car-racing test accuracy of 74.0% (started from 19.0%).
>
> [b] Weight Agnostic Neural Networks https://arxiv.org/pdf/1906.04358.pdf
>
> [c] AutoML-Zero: Evolving Machine Learning Algorithms From Scratch https://arxiv.org/pdf/2003.03384.pdf

---

### Author Response · Authors · 2020-11-25
**Global Comment Addressing Core Contributions**

Thank you to all of the reviewers for their time both reading and providing feedback on our paper, we greatly appreciate it!

To summarize, we believe our main contributions with this work are:

              (1) The first demonstration of artificial, architecture agnostic neural networks (AAAN), that retain biological plausibility. AAAN refers to a family of networks that are functionally similar but architecturally distinct.

              (2) The demonstration that network families with common architectural properties share similar accuracies and structural properties.

              (3) The demonstration that moving in the architecture manifold improves performance with both randomly initialized and backpropagation trained models.

We view our “stochastic search and succeed” algorithm as a tool to facilitate the exploration of the principles outlined above while maintaining generalizability and robustness. We view our work as studying a biologically plausible strategy for studying neural network learning, not as a core algorithms contribution.

---

### Decision · Program_Chairs · 2021-01-07
**Final Decision**

**Decision:**

Reject

**Comment:**

This paper explores methods for pruning binary neural networks. The authors provide algorithms for developing sparse binary networks that perform okay on some basic ML benchmarks. They frame this as providing insights into synaptic pruning in the brain, and potentially providing a method for more efficient edge computing in the future.

All four reviews placed the paper below the acceptance threshold. The reviewers noted that the paper was hard to follow in several places and were unsure as to the motivations. The authors attempted to address these concerns in their replies, but the Area Chair felt that these were insufficient.

As well, the Area Chair notes that some of the claimed contributions of the paper are questionable. Specifically:

(1) The claim that there is anything biologically plausible about the algorithms presented here is very suspect. The brain cannot use a search and test system for synaptic pruning like the algorithms proposed here. Thus, it is unclear how this paper provides any insight for neuroscience. In fact, the authors do not even really try to provide any neuroscience insights in the results or discussion. Moreover, they don't actually appear to use any neuroscience insights to develop their algorithms, other than the stochasticity of the pruning (though note: it is not actually clear in neuroscience data whether pruning is stochastic). Given the ultimately very poor performance on ML tasks, the paper doesn't seem to provide anything particularly useful for application in ML either.

(2) The claim that the provide, "The demonstration that network families with common architectural properties share similar accuracies and structural properties." is odd. Surely this is the null hypothesis anyone would have about ANNs? It would be surprising if networks with common connectivity profiles (which is what the authors mean by "architecture") didn't share similar performance!

(3) The claim that searching in architecture space like this leads to "architecture agnostic networks" is odd... As noted by Reviewer 2, the authors are really just specifying algorithms for sparsifying binary neural networks, which they frame as being "architecture agnosticism" according to a rather strained definition. There are other ways of approaching the sparsification of neural networks, and of doing architecture optimization, but the paper is not framed as contributing to this literature.

Altogether, given these considerations, and the four reviews, a "Reject" decision was delivered.